# Extracellular-Vesicle-Associated UBE2NL and HIST2H3PS2 Promote Tumor Aggressiveness and Metastasis in Gynecologic Cancer

**DOI:** 10.3390/ijms26104833

**Published:** 2025-05-18

**Authors:** Chih-Ming Ho, Ting-Lin Yen, Tzu-Hao Chang, Shih-Hung Huang

**Affiliations:** 1Gynecologic Cancer Center, Department of Obstetrics and Gynecology, Cathay General Hospital, Taipei 106, Taiwan; 2School of Medicine, Fu Jen Catholic University, Hsinchuang, New Taipei City 242, Taiwan; 3Department of Medical Research, Cathay General Hospital, Sijhih, New Taipei City 221, Taiwan; d119096015@tmu.edu.tw; 4School of Medicine, Taipei Medical University, Taipei 110, Taiwan; 5Graduate Institute of Biomedical Informatics, Taipei Medical University, Taipei 110, Taiwan; kevinchang@tmu.edu.tw; 6Department of Pathology, Cathay General Hospital, Taipei 106, Taiwan; ja68@cgh.org.tw

**Keywords:** extracellular vesicles, UBE2NL, HIST2H3PS2, ovarian cancer, metastasis, invasion, tumor microenvironment

## Abstract

Extracellular vesicles (EVs) play pivotal roles in tumor progression and metastasis by mediating intercellular communication within the tumor microenvironment. In this study, we identified two novel EX cargo proteins—UBE2NL and HIST2H3PS2—derived from highly aggressive epithelial ovarian cancer (EOC) cells and mesenchymal-type ovarian stromal progenitor cells (MSC-OCSPCs) but absent in less aggressive SKOV3 cells. Quantitative proteomic profiling via LC-MS/MS and TCGA-integrated analysis revealed that high expression of these genes correlated with advanced tumor stages and poor overall survival in EOC, and high HIST2H3PS2 expression predicted poor survival in endometrial cancer (EC). Functionally, UBE2NL and HIST2H3PS2 overexpression promoted EOC cell invasiveness, which was further enhanced by EX-mediated autocrine and paracrine effects. In contrast, the knockdown of UBE2NL reduced cell invasiveness and prolonged mouse survival in vivo. Moreover, HIST2H3PS2-enriched EXs significantly increased peritoneal dissemination and ascites in murine models. These findings suggest that EX-packaged UBE2NL and HIST2H3PS2 drive tumor aggressiveness and metastasis in gynecologic cancers, highlighting their potential as prognostic biomarkers and therapeutic targets.

## 1. Introduction

Ovarian cancer remains one of the most lethal gynecological malignancies worldwide due to the lack of early symptoms and reliable screening tools [1]. As a result, most patients are diagnosed at an advanced stage, making complete surgical resection difficult and contributing to poor overall prognosis [2]. Advanced epithelial ovarian cancer (EOC) often presents with extensive peritoneal dissemination, residual disease following debulking surgery, and resistance to chemotherapy [2,3,4]. Notably, more than one-third of patients develop malignant ascites, which is a fundamental part of chemo-resistant and recurrent disease [5,6]. These clinical challenges underscore the urgent need for novel biomarkers and therapeutic strategies.

Similarly, endometrial cancer (EC), the most prevalent gynecologic malignancy, primarily affects postmenopausal women. However, its incidence has increased in younger populations due to risk factors such as obesity [7,8]. While type I (estrogen-dependent) EC is usually diagnosed early and associated with favorable outcomes, type II EC is more aggressive and carries a poorer prognosis [8]. Additionally, disparities in survival outcomes remain across socioeconomic and racial groups, further emphasizing the need for personalized and targeted treatment strategies [9].

Extracellular vesicles (EVs), including exosomes (EXs) and microvesicles, have emerged as key mediators of tumor progression, metastasis, chemotherapy resistance, and immune modulation [10,11]. Cancer cells actively release EVs, which participate in remodeling the tumor microenvironment and facilitate communication between cancer cells and surrounding stromal or immune cells [12]. These vesicles transport a wide range of bioactive molecules, such as microRNAs, mRNAs, and proteins [13]. Due to their stability and presence in biological fluids, EXs are attractive candidates as non-invasive biomarkers for cancer detection and prognosis [14].

In this study, we aimed to explore the role of EXs derived from aggressive ovarian cancer cells in modulating tumor aggressiveness. Using quantitative proteomics via liquid chromatography–mass spectrometry (LC-MS/MS), we analyzed protein expression profiles in ES2 cells, ES2 cells treated with ES2-derived EXs, and ascites-derived mesenchymal-type ovarian carcinoma stromal progenitor cells (MSC-OCSPCs) treated with ES2 EXs. This allowed us to investigate the autocrine and paracrine effects of EXs in shaping tumor behavior and stromal cell interaction. By integrating proteomic data with clinical survival data from The Cancer Genome Atlas (TCGA), we identified UBE2NL and HIST2H3PS2 as two novel EX cargo proteins significantly enriched in aggressive cells and associated with poor progression-free and overall survival in EOC.

UBE2NL, a ubiquitin-conjugating enzyme-like protein, is structurally related to UBE2N, a protein involved in DNA repair, inflammatory signaling, and cancer progression [15]. It has a link to poor prognosis in cancers such as breast cancer with metastatic lung involvement [16]. UBE2N’s involvement in DNA damage response and K63-linked ubiquitination is crucial for DNA repair protein recruitment, and its low expression or overexpression is associated with chemotherapy resistance and poor outcomes in some cancers [17,18,19]. While UBE2NL is less well-characterized, it likely participates in K63-linked polyubiquitination, particularly through interaction with the co-factor UBE2V2. UBE2V2 is a non-catalytic E2 variant implicated in protein degradation, cell-cycle regulation, and oncogenic signaling [20,21,22,23]. Emerging evidence suggests that disrupting UBE2NL–UBE2V2 complexes may be a promising therapeutic strategy to block tumor-promoting ubiquitination cascades [20].

HIST2H3PS2 is a pseudogene of histone H3 with limited functional characterization in cancer [24]. However, recent studies indicate that it is hypermethylated in endometrial cancer tissues compared to normal endometrium, suggesting a potential role in epigenetic regulation [25]. HIST2H3PS2 also displays tissue-enhanced expression in the ovary, further implicating it in gynecologic tumor biology [26]. Despite its unclear mechanistic role, its elevated expression in aggressive ovarian and endometrial cancers suggests a potential oncogenic function worth further investigation.

Together, the identification of UBE2NL and HIST2H3PS2 in EXs derived from aggressive ovarian cancer cells provides new insights into the molecular underpinnings of gynecologic cancer progression and highlights their potential as prognostic biomarkers and therapeutic targets.

## 2. Results

### 2.1. LC-MS/MS Proteomic Analysis and Integration with TCGA Data to Identify Key EX-Associated Proteins

Our previous studies demonstrated the isolation and characterization of EOC and ascites-derived cell lines and extracellular exosomes (EXs) [27,28,29]. The EXs derived from epithelial ovarian cancer (EOC) cells exert both autocrine and paracrine effects, enhancing the invasiveness of both EOC cells and mesenchymal-type ovarian carcinoma stromal progenitor cells (MSC-OCSPCs) in the tumor microenvironment. Specifically, EXs from aggressive ES2 cells significantly increased the invasive capacity of ES2 cells and MSC-OCSPCs, while EXs from less aggressive SKOV3 cells had a reduced effect [29].

We performed proteomic profiling using liquid chromatography–mass spectrometry (LC-MS/MS) to identify novel EX-associated proteins that promote tumor aggressiveness. The differentially expressed proteins (DEPs) were selected based on strict filtering criteria, including a false discovery rate (FDR) of <0.05 and a fold change of >2. To evaluate both autocrine and paracrine effects of EXs, we defined three comparison groups: (1) ES2 cells treated with ES2 EXs versus untreated ES2 cells, (2) MSC-OCSPCs treated with ES2 EXs versus untreated MSC-OCSPCs, and (3) MSC-OCSPCs treated with ES2 EXs versus ES2 cells treated with ES2 EXs.

Three genes—UBE2NL, HIST2H3PS2, and SLC25A31—were identified as significantly enriched in EX-treated aggressive cell groups and were further analyzed for clinical relevance using survival data from The Cancer Genome Atlas (TCGA). Multivariate Cox regression analysis revealed that higher expression levels of UBE2NL, HIST2H3PS2, and SLC25A31 were associated with poorer progression-free survival (PFS), with exp(coef) values of 3.007, 3.709, and 61.374, respectively (Appendix A).

Further validation was conducted using Kaplan–Meier survival analysis on TCGA datasets for all subtypes and stages of ovarian cancer. High expression levels of UBE2NL and HIST2H3PS2 were significantly associated with poorer PFS and overall survival (OS) compared to the low-expression groups. Specifically, hazard ratios for OS in all-stage EOC patients were 1.48 (95% CI: 1.13–1.93) for UBE2NL and 1.85 (95% CI: 1.38–2.46) for HIST2H3PS2 (Figure 1a,b). When both genes of UBE2NL and HIST2H3PS2 were considered, progression-free survival and overall survival were better in patients with low expressions of both genes than in those with high expressions (*p* = 0.017 for PFS, *p* = 0.022 for OS) (Figure 1c,d).

We next asked whether the expression of UBE2NL and HIST2H3PS2 also affected clinical outcomes in endometrial cancer (EC) patients. Using TCGA survival data for EC, we found that high expression of HIST2H3PS2 was significantly associated with poorer PFS and OS in all-stage EC patients based on log-rank tests. Specifically, the high-expression group had markedly reduced survival compared to the low-expression group (Figure 2a,b). In contrast, UBE2NL expression levels did not affect survival outcomes in EC patients.

### 2.2. mRNA Expression Levels of UBE2NL and HIST2H3PS2 in Benign and Malignant Ovarian Tissues

To investigate the clinical relevance of UBE2NL and HIST2H3PS2 expression in ovarian cancer progression, we examined their mRNA expression levels in tissue samples obtained from patients at Cathay General Hospital (CGH), including benign ovarian cysts (n = 10), early-stage ovarian cancer (n = 21), and advanced-stage ovarian cancer (n = 35). Statistical analysis was performed using the Kruskal–Wallis test.

The mean mRNA expression levels of UBE2NL showed a significant increase across disease stages. Specifically, UBE2NL expression was significantly higher in advanced-stage ovarian cancer tissues compared to early-stage cancer and benign ovarian cysts (mean ± SD: 0.1225 ± 0.2508 vs. 0.0862 ± 0.1662 vs. 0.0037 ± 0.0069; *p* < 0.001, by Kruskal–Wallis test).

For HIST2H3PS2, the mean expression level was also elevated in advanced-stage ovarian cancer and early-stage ovarian cancer compared to benign cysts (mean ± SD: 0.0002 ± 0.0003 vs. 0.0003 ± 0.0003 vs. 0.0001 ± 0.0001; *p* > 0.05, by Kruskal–Wallis test). The dot plots illustrate the distribution of UBE2NL and HIST2H3PS2 mRNA in ovarian tissues among benign ovarian cysts, early-stage ovarian cancer, and advanced-stage ovarian cancers in Figure 3.

These results suggest that increased expression of both UBE2NL and HIST2H3PS2 is associated with disease progression and may serve as potential molecular markers to differentiate benign, early-stage, and advanced-stage ovarian tumors.

### 2.3. Isolation and Characterization of UBE2NL- and HIST2H3PS2-Enriched Exosomes in EOC Cell Lines

We established a standardized operating protocol for EX isolation and analysis in our laboratory to investigate the roles of UBE2NL and HIST2H3PS2 in extracellular vesicle (EX) biology. Exosomes were isolated from EOC cell culture supernatants using the ExoQuick-TC™ kit and subsequently characterized by nanoparticle tracking analysis (NTA), transmission electron microscopy (TEM), and Western blotting for established EX markers.

We verified the size and concentration of EXs using NTA. The average particle sizes of exosomes derived from SKOV3 and SKOV3-overexpressing HIST2H3PS2 (SKOV3/HIST2H3PS2) cells were 143.0 nm and 131.2 nm, respectively. The corresponding EX concentrations were 3.3 × 10^7^ particles/mL and 9.5 × 10^7^ particles/mL (Figure 4a).

Morphological evaluation using TEM confirmed the characteristic cup-shaped structures of exosomes isolated from SKOV3 and SKOV3/HIST2H3PS2 cell lines (Figure 4b). To further validate the exosome identity, we performed Western blotting to detect the expression of CD9, a well-established EX marker. CD9 was expressed in EXs derived from SKOV3, SKOV3-overexpressing UBE2NL, and SKOV3-overexpressing HIST2H3PS2 cells. Notably, CD9 was not detected in the corresponding cellular lysates, confirming the specificity of the marker to EX fractions. The UBE2NL expression was more pronounced in exosomes derived from SKOV3-over UBE2NL cells than in SKOV3 cells (Figure 4c).

### 2.4. Expression of UBE2NL and HIST2H3PS2 in EOC Cells and Their Exosomes

UBE2NL expression was observed in aggressive cell types and EXs, including ES2, paclitaxel-resistant ES2TR, sphere-forming ES2TS and ES2TR-TS, SKOV3, and MSC-OCSPCs. In contrast, UBE2NL was not detected in EXs from the less aggressive SKOV3 cells (Figure 5a). Interestingly, although ES2 cells are generally considered more aggressive than SKOV3 cells, the basal mRNA expression level of UBE2NL was lower in ES2 cells compared to SKOV3 cells. This discrepancy may be due to intrinsic molecular differences between the two cell lines from distinct histological subtypes of ovarian cancer. Moreover, the basal expression levels may not fully reflect the functional impact of UBE2NL on tumor aggressiveness. In our subsequent functional assays, we observed that overexpression or exosome-mediated transfer of UBE2NL markedly enhanced aggressive behaviors in EOC cells, supporting its role in promoting malignancy beyond basal expression levels alone. Notably, strong expression of UBE2NL in EXs derived from ES2TR and MSC-OCSPCs suggests a potential role in paclitaxel resistance.

Similarly, HIST2H3PS2 was expressed in aggressive cell lines and their EXs, including ES2, ES2TR, ES2TS, ES2TRTS, SKOV3, and MSC-OCSPCs. However, HIST2H3PS2 was absent in SKOV3 cells and EXs (Figure 5b). The high expression of UBE2NL and HIST2H3PS2 in MSC-OCSPCs represents their involvement in the tumor microenvironment.

To further validate the exosome expression of these genes, we overexpressed UBE2NL and HIST2H3PS2 in SKOV3 cells (SKOV3/UBE2NL and SKOV3/HIST2H3PS2) and performed knockdown of UBE2NL in ES2 cells (ES2/shUBE2NL). Quantitative RT-PCR analysis revealed that UBE2NL and HIST2H3PS2 mRNA levels were significantly higher in SKOV3/UBE2NL and SKOV3/HIST2H3PS2 cells and their derived EXs compared to wild-type SKOV3 cells and EXs (Figure 6a,b). Furthermore, the expression of UBE2NL was markedly lower in ES2/shUBE2NL cells and their EXs compared to wild-type ES2 counterparts (Figure 6a).

These results confirm that UBE2NL and HIST2H3PS2 are expressed within aggressive EOC cells and secreted into the extracellular space via exosomes.

### 2.5. Functional Characterization of UBE2NL and HIST2H3PS2

We conducted cell viability and invasion assays using overexpression and knockdown models to investigate the functional roles of UBE2NL and HIST2H3PS2 in EOC cell proliferation and invasiveness.

Cell viability was assessed using the CCK-8 assay. SKOV3 cells overexpressing UBE2NL (SKOV3/UBE2NL) or HIST2H3PS2 (SKOV3/HIST2H3PS2) demonstrated significantly higher viability compared to wild-type SKOV3 cells (Figure 7a,b). Conversely, knockdown of UBE2NL in SKOV3 cells (SKOV3/shUBE2NL) resulted in a significant reduction in cell viability compared to SKOV3 controls (Figure 7c).

Next, we evaluated the invasive potential of these modified cell lines. Transwell invasion assays revealed that SKOV3 cells overexpressing UBE2NL or HIST2H3PS2 exhibited significantly greater invasive ability compared to SKOV3 cells overexpressing luciferase (SKOV3/Luc) used as a control (Figure 8a,b).

In contrast, the invasive ability of ES2 cells was markedly reduced when UBE2NL was knocked down. ES2/shUBE2NL cells exhibited significantly lower invasion than wild-type ES2 cells (Figure 9).

### 2.6. Functional Characterization of UBE2NL- and HIST2H3PS2-Enriched Exosomes

To assess the functional impact of UBE2NL- and HIST2H3PS2-enriched exosomes (EXs) on tumor cell invasion, we conducted transwell invasion assays using SKOV3 cells with various EX treatments.

In the UBE2NL-related experiments, SKOV3 cells treated with SKOV3-derived EXs or SKOV3/UBE2NL-derived EXs showed significantly enhanced invasion compared to untreated SKOV3 cells. Moreover, SKOV3/UBE2NL cells co-treated with SKOV3/UBE2NL-derived EXs exhibited the highest level of invasion, indicating both autocrine and paracrine pro-invasive effects of UBE2NL-enriched EXs (Figure 10a).

Similarly, SKOV3 cells treated with SKOV3-derived EXs or SKOV3/HIST2H3PS2-derived EXs displayed significantly increased invasion capacity compared to control SKOV3 cells. SKOV3/HIST2H3PS2 cells supplemented with their EXs demonstrated the most pronounced invasive potential (Figure 10b).

These results indicate that UBE2NL and HIST2H3PS2, when packaged in exosomes, promote invasion through both autocrine and paracrine mechanisms, reinforcing their role in driving tumor aggressiveness.

In addition, we tested whether exosomes from UBE2NL knockdown cells suppress invasion. ES2 cells or ES2/shUBE2NL cells were treated with EXs derived from ES2/shUBE2NL cells. In both cases, the invasion ability was significantly reduced compared to those cells treated with ES2-derived EXs (Figure 11). This suggests that UBE2NL knockdown impairs the pro-invasive function of ES2-derived exosomes.

### 2.7. In Vivo Validation of UBE2NL and HIST2H3PS2 Roles in Tumor Progression

The in vivo experiment of IP injection with ES2 and ES2/shUBE2NL cells.

During an in vivo experiment, 8 out of 10 null mice (BALB/cAnN.Cg-Foxn1nu/CrlNarl) expired after IP injection of 1 × 10^6^ ES2 cells, compared to 2 out of 10 mice that expired after IP injection of 1 × 10^6^ ES2 (shUBE2NL) on the 32nd day (Figure 12a). The survival rate was significantly higher in ES2/shUBE2NL than in ES2 cells by Kaplan–Meier (*p* < 0.05, Log-rank test). However, the body weights before and after IP injections of cells were not significantly different (Figure 12b).

In addition, the average number of disseminated tumor nodules in the peritoneal cavity was significantly lower in mice (n = 5) injected with 1 × 10^6^ ES2/shUBE2NL cells compared to those injected with ES2 cells (Figure 13a,b). This finding further supports the role of UBE2NL in promoting intraperitoneal tumor dissemination.

### 2.8. In Vivo Validation of UBE2NL- and HIST2H3PS2-Enriched Exosomes in Mediating Tumor Progression

To assess whether exosomes enriched in HIST2H3PS2 contribute to tumor aggressiveness in vivo, we conducted intraperitoneal (IP) injection experiments using SKOV3 or SKOV3/HIST2H3PS2 cells, with or without their respective EXs.

Mice injected with 1 × 10^6^ SKOV3/HIST2H3PS2 cells developed significantly more severe ascites than mice injected with 1 × 10^6^ SKOV3 cells (Figure 14). This observation suggests that HIST2H3PS2 overexpression promotes peritoneal inflammation and fluid accumulation, indicative of advanced disease status.

Furthermore, the average number of disseminated tumor nodules in the peritoneal cavity was higher in mice (n = 5) injected with 1 × 10^6^ SKOV3/HIST2H3PS2 cells compared to those injected with 1 × 10^6^ SKOV3-overexpressing luciferase (SKOV3/Luc) cells (Figure 15a and Figure 16a).

To determine the role of exosomes in mediating tumor spread, we injected mice with SKOV3/Luc cells alone or in combination with exosomes. Mice receiving SKOV3/Luc cells plus EXs had significantly more disseminated tumors than those injected with SKOV3/Luc cells alone (Figure 15a,b and Figure 16c).

Similarly, mice injected with SKOV3/HIST2H3PS2 cells and their derived EXs showed a significantly higher number of disseminated tumor nodules than those injected with SKOV3/HIST2H3PS2 cells alone (Figure 15a,b and Figure 16b).

Moreover, a comparison between mice receiving SKOV3/HIST2H3PS2 cells + 10 μg EXs and those receiving SKOV3/Luc cells + 10 μg EXs revealed significantly higher tumor dissemination in the former group (Figure 15b and Figure 16d). These findings demonstrate that HIST2H3PS2-enriched exosomes further potentiate tumor dissemination in vivo.

### 2.9. Effects of Exosome Inhibition in ES2 Cells

To assess whether targeting exosome secretion affects tumor aggressiveness, we treated ES2 cells with known EX inhibitors (Figure 17a). Treatment with 10 μM GW4869 (a neutral sphingomyelinase inhibitor) or (Figure 17b) 500 nM rapamycin (an autophagy activator) significantly reduced the level of autocrine EXs secreted by ES2 cells.

These results suggest that inhibition of exosome biogenesis or secretion can potentially attenuate the autocrine signaling loop in aggressive EOC cells and may represent a promising therapeutic approach to reduce tumor invasiveness.

## 3. Discussion

This study first identified that high expressions of UBE2NL and HIST2H3PS2 were associated with poor survival outcomes in epithelial ovarian cancer (EOC) patients using TCGA data, and we validated these findings in clinical samples across all subtypes of EOC. Expression levels of both genes were significantly elevated in the advanced-stage compared to the early-stage disease. Notably, their expression was also high in paclitaxel-resistant ES2 cells and ascites-derived mesenchymal ovarian cancer stromal progenitor cells (MSC-OCSPCs), suggesting a link with aggressive tumor behavior. Furthermore, high expression of HIST2H3PS2 was associated with poor progression-free and overall survival in endometrial cancer (EC) patients, underscoring its broader relevance in gynecologic malignancies. These findings suggest that UBE2NL and HIST2H3PS2 are potential prognostic biomarkers and therapeutic targets in gynecologic cancers.

Our findings are consistent with previous studies demonstrating that Ubiquitin-conjugating enzymes such as UBE2N are increasingly recognized for their roles in DNA repair, immune signaling, and oncogenesis [15,31]. UBE2N, in complex with UBE2V2, modulates DNA repair proteins like PCNA and p53 through K63-linked ubiquitination, which is crucial for maintaining genomic stability [32,33]. Overexpression of UBE2N has been associated with poor prognosis, chemotherapy resistance, and enhanced NF-κB signaling in various cancers [34,35]. Although UBE2NL is less studied, its structural similarity to UBE2N and ability to dimerize with E2V2 suggest a potential role in similar pathways, particularly those regulating protein degradation and cell-cycle progression [20,36].

Similarly, our observation is that HIST2H3PS2, a histone H3 pseudogene, may contribute to cancer progression through its involvement in chromatin remodeling and epigenetic regulation [37]. Epigenetic alterations such as histone acetylation and methylation are involved in ovarian cancer progression and chemotherapy drug resistance through mechanisms involving histone deacetylases (HDACs) and dysregulated oncogene or tumor suppressor expression [38,39]. Histone deacetylase (HDAC) inhibitors have shown efficacy in reversing chemotherapy resistance in ovarian cancer models, underscoring the importance of epigenetic regulators as therapeutic targets [39].

Using the SKOV3 cell model, we demonstrated that overexpression of UBE2NL and HIST2H3PS2 significantly enhanced cell invasion. This effect was amplified when exosomes (EXs) derived from these overexpressing cells were present, highlighting an important role for exosome-mediated autocrine and paracrine signaling in promoting tumor aggressiveness. In contrast, the knockdown of UBE2NL or inhibition of EX secretion markedly reduced cellular invasiveness, confirming the functional contribution of these genes and their EXs.

In vivo studies further supported these in vitro findings. Mice injected with ES2/shUBE2NL cells had higher survival rates and fewer disseminated tumors than those injected with wild-type ES2 cells. Likewise, mice injected with SKOV3/HIST2H3PS2 cells developed more extensive tumor dissemination and ascites. Importantly, tumor burden was significantly greater in mice injected with SKOV3/HIST2H3PS2 cells plus their corresponding EXs than in control groups. These results suggest that HIST2H3PS2-enriched EXs are critical mediators of tumor progression.

Although our study focused on the local effects of extracellular vesicles (EVs) within the tumor microenvironment, it did not assess systemic EV biodistribution. Nevertheless, our intraperitoneal injection models demonstrated that HIST2H3PS2-enriched EVs enhanced tumor dissemination, ascites formation, and metastatic potential. Functional validation using EV-depleted or heat-inactivated EVs may further confirm their pro-tumorigenic effects.

One limitation of this study is the absence of luciferase-labeled cells and whole-body imaging, which may have improved metastasis quantification. However, we carefully evaluated tumor burden through direct gross and histologic inspection, following validated protocols from prior studies [40].

Our research presents several novel insights compared to previous studies. While the oncogenic roles of UBE2N and tumor-derived EVs have been previously reported [11,35], our work is the first to demonstrate a functional role for UBE2NL and HIST2H3PS2, particularly in the context of EV-mediated tumor progression in gynecologic cancers. Furthermore, the role of histone pseudogenes in exosome biology remains unexplored, highlighting the novelty of our findings.

### Conclusions and Future Perspectives

In conclusion, this study identifies UBE2NL and HIST2H3PS2 as novel contributors to tumor aggressiveness and metastasis in epithelial ovarian and endometrial cancer. Both genes promote invasive behavior in vitro and facilitate tumor dissemination in vivo, effects enhanced by their respective exosomes. Our findings highlight the significance of EV-mediated intercellular communication in gynecologic cancer progression.

Future investigations should focus on several key areas. First, mechanistic studies are needed to elucidate the molecular signaling pathways downstream of UBE2NL and HIST2H3PS2, particularly their influence on NF-κB activation, DNA damage repair, and chromatin remodeling. Second, therapeutic strategies targeting these genes, their interacting partners, or their secreted EVs should be verified. Inhibitors of ubiquitination pathways or EV biogenesis may provide novel treatment avenues. Third, biomarker development using UBE2NL, HIST2H3PS2, or their EV-associated cargo could improve patient risk stratification and monitoring of treatment response. Fourth, detailed EV cargo profiling via proteomic and transcriptomic analyses could reveal specific oncogenic molecules contributing to metastasis. Finally, combination therapies integrating HDAC inhibitors and EV-targeted approaches may provide synergistic anti-tumor effects. Targeting UBE2NL, HIST2H3PS2, and their EV-mediated effects may represent a promising new frontier in treating aggressive gynecologic cancers.

## 4. Materials and Methods

### 4.1. Sample Collection

Informed consent for sample collection was obtained from all participants involved in the study. Ovarian cancer tissues and ascites samples, obtained during surgery or for symptom relief from patients with primary or recurrent ovarian cancer, were immediately transported to the laboratory for further processing. Isolation and culture of ovarian cancer stromal progenitor cells (OCSPCs) from these samples followed established protocols.

### 4.2. Cell Lines and Culture

ES2 and SKOV3 cell lines were from the American Type Culture Collection (ATCC), Manassas, VA, USA. ES2 is a cell line exhibiting fibroblast-like morphology with clear cell carcinoma features. SKOV3 is a cell line displaying epithelial morphology with gynecologic high-grade carcinoma characteristics derived from uterine or ovarian serous carcinoma. Cells were cultured in McCoy’s 5A medium supplemented with 10% FBS and maintained at 37 °C in a humidified atmosphere with 5% CO_2_. Cells were seeded at a density of 2 × 10^5^ cells per 10 cm dish in 10 mL of complete medium and passaged every 2–3 days upon reaching 80–90% confluence. Paclitaxel-resistant ES2 cells (ES2TR) were developed by continuous exposure to increasing concentrations of paclitaxel, with a final selection concentration of 160 nM.

### 4.3. Spheroid Formation of Ovarian Cancer Stem-like Cells

ES2, ES2TR160, and ascites-derived cells from EOC patients were cultured under spheroid-inducing conditions in DMEM/F12 medium supplemented with 20 ng/mL bFGF, 20 ng/mL EGF, 10 ng/mL IGF, and 2% B27. Single cells (1 × 10^5^ cells/mL) were seeded in ultra-low attachment plates. Spheres formed after seven days were counted using an Olympus light microscope, Olympus Corporation, Tokyo, Japan, and spheroids were harvested after 14 days for flow cytometry.

### 4.4. Exosome Isolation

To collect exosomes, we seeded ES2, ES2TR, and SKOV3 cells at a density of 2 × 10^6^ cells per 10 cm dish in 10 mL of complete culture medium. Once the cells reached approximately 95% confluence, the medium was replaced with serum-free McCoy’s 5A medium and incubated for 24 h. Conditioned media were harvested and centrifuged at 3000× *g* for 15 min at 4 °C to remove cells and cellular debris, eliminating larger EVs and apoptotic bodies. The clarified supernatants were transferred to a fresh tube and mixed with ExoQuick-TC™ solution (System Biosciences, Palo Alto, CA, USA) at a 1:5 (*v*/*v*) ratio according to the manufacturer’s instructions. The mixture was gently inverted to mix and incubated overnight at 4 °C. Exosomes were subsequently pelleted by centrifugation at 1500× *g* for 30 min, followed by an additional centrifugation at 1500× *g* for 5 min to remove residual ExoQuick-TC reagent. The final exosome pellet was gently resuspended in 100 μL of sterile 1× PBS and stored at −80 °C until further use.

### 4.5. Nanoparticle Tracking Analysis (NTA)

The method was described in Reference [29]. Briefly, purified exosomes resuspended in 100 μL of 0.22 μm filtered PBS were analyzed using the NanoSight LM10 (NanoSight, Salisbury, UK) with a 404 nm laser. The Nanoparticle Tracking Analysis software (v2.3) processed 60 s video data to determine vesicle size and concentration metrics.

### 4.6. Transmission Electron Microscopy (TEM)

The isolated exosomes underwent a process where they were placed on a glow-discharged carbon-coated EM grid, stained with 1% uranyl formate, and dried at room temperature overnight. Subsequently, electron microscopy (FEI Tecnai G2 F20 S-TWIN, FEI Company, Hillsboro, OR, USA) at 120 kV was used to examine the grid, and the images were captured during the process.

### 4.7. Western Blot Analysis

Approximately 30 µg of exosome protein (quantified by Bradford assay), corresponding to approximately 8 × 10^9^ exosome particles, as quantified by nanoparticle tracking analysis (NTA) per lane for Western blot analysis of exosomes. Cells were lysed in phosphate-buffered saline (PBS) containing 1% Triton X-100 using an ultrasonic disruptor. The resulting lysates were subjected to 12.5% SDS-PAGE and transferred onto a polyvinylidene fluoride (PVDF) membrane (NEN). Following the transfer, membranes were blocked with tris-buffered saline supplemented with 0.2% Tween 20 and 1% I-block (NEN) before incubation with polyclonal antibodies for 1 h. To normalize the signals, a purified rabbit anti-human GAPDH polyclonal antibody (Santa Cruz Biotechnology, Inc., Dallas, TX, USA) was applied alongside the anti-CD9 antibody (Cell Signaling, Danvers, MA, USA). After thorough washing, an alkaline phosphatase-conjugated anti-rabbit secondary antibody (Vector Laboratories, Burlingame, CA, USA) was added. Finally, bound antibodies were detected using nitroblue tetrazolium/5-bromo-4-chloro-3-indolyl phosphate chromogen.

### 4.8. Exosome Quantification

The protein concentration of exosome lysates was determined using a Bradford protein assay (Sigma, B6916, St. Louis, MO, USA). Briefly, exosome samples were lysed, diluted in sterile 1× PBS, and mixed with Bradford reagent according to the manufacturer’s instructions. After incubation at room temperature, we measured the absorbance at 595 nm. We prepared the standard curve using bovine serum albumin (BSA), and the protein concentration of exosome lysates was calculated and expressed as mg/mL.

### 4.9. Gene Knockdown and Overexpression of UBE2NL and HIST2H3PS2

We purchased the UBE2NL and HIST2H3PS2 knockdown and overexpression plasmids from the National RNAi Core Facility at Academia Sinica in Taiwan. For UBE2NL knockdown in ES2 cells, we used human pLKO TRC005 lentiviral shRNA plasmids (TRCN0000365724) targeting the sequences 5′-ACCCAGACATCTTCAGTTATT-3′. After infection, we treated with puromycin (2 µg/mL) to select the infected cells. For UBE2NL and HIST2H3PS2 overexpression in SKOV3 cells, we used the cDNA fragment encoding UBE2NL and HIST2H3PS2 cloned into pLAS3w.Ppuro in NheI and PmeI sites of a puromycin-resistant lentiviral vector from the National RNAi Core Facility at Academia Sinica in Taiwan.

### 4.10. qPCR for UBE2NL and HIST2H3PS2

RNA was isolated by RNA purification kits from MACHEREY-NAGEL and stored at −80 °C before use. We evaluated the RNA quantity and quality by spectrophotometric analysis. We subjected one microgram of total RNA to synthesize the first-strand cDNA using the High-Capacity cDNA Reverse Transcription Kit (Thermofisher, Waltham, MA, USA). We performed one microliter of the reverse transcription product used for PCR amplification in a final 10-μL reaction with the following conditions: 95 °C for 2 min, followed by 60 cycles of 95 °C for 10 s, 60 °C for 20 s, and a final cooling step of 40 °C for 30 s. We amplified the genes with primers designed by Roche, Basel, Switzerland as shown below: We used GAPDH as an internal control, and the software analysis was applied using Roche LightCycler Software 4.05. The PCR product was detected using the specific primers of UBE2NL forward, 5′-GCAGAACCAGATGAAAGCAAC GC -3′ and reverse, 5′-GGGCTGCCATTGGGTATTCTTC-3′; and HIST2H3PS2 forward, 5′-GCTGTTCGAAGACACGAACC-3′ and reverse, 5′-CGGCTGACCAACTGGATGT-3′; GAPDH forward, 5′-GAGTCAACGGATTTG GTCGT-3′ and reverse; 5′-TTGATTTTGGAGGGATCTCG-3′ on a StepOne Plus RealTime PCR System (Applied Biosystems; Thermo Fisher Scientific, Inc., Austin, TX, USA) using the SYBR Green PCR Kit (QIAGEN GmbH, Hilden, Germany). We calculated the relative mRNA expression levels using the 2^−∆∆Ct^ method, and GAPDH served as an internal control.

### 4.11. Cell Proliferation and Viability (CCK-8 Assay)

The cell proliferation and viability of SKOV3, SKOV3-overluciferase, SKOV3/UBE2NL, SKOV3/HISH2T3PS2, ES2, ES2 vector, ES2/shUBE2NL cells were assessed by CCK8 (DOJINDO, Kamimashiki-gun, Japan) assay. Briefly, cells (10,000 cells/well) in a 96-well plate were exposed to the culture medium and served as a control. CCK8 was added to the cells at the final 0.5 mg/mL and incubated at 37 °C for two hours. At the end of incubation, the optical density was measured at 450 nm using a universal microplate reader ElX 800 (Bio-Tek Instruments, Winooski, VT, USA).

### 4.12. LC-MS/MS Analysis

We used the reduced condition media with 10 mM dithiothreitol, alkylated with 50 mM iodoacetamide, and digested with Lys-C and trypsin. The digested peptides were labeled with isotopic formaldehyde (13CD2O, heavy labeled) and formaldehyde (CH2O, light labeled), respectively. Equal amounts of the heavy and light labeled peptides were mixed and desalted with StageTips with Empore™ SDB-CX disc membrane (3M, St. Paul, MN, USA). NanoLC-MS/MS analyses. The peptides were analyzed using nanoLC-MS/MS on a Dionex 3000 RSLC nanosystem (Thermo Fisher Scientific) and online coupled to an LTQ Orbitrap XL mass spectrometer (Thermo Fisher Scientific). The supernatant was dried using SpeedVac. Redissolved peptides with 0.5% acetic acid and 2% acetonitrile (ACN) were loaded onto an in-house-prepared 100 μm × 15 cm tip column, packed with 3 μm ReproSil-Pur 120 C18-AQ reverse-phase beads, and eluted at a flow rate of 500 nL/min. The mobile phases used for nanoLC will be 0.5% acetic acid in water (buffer A) and a mixture of 0.5% acetic acid and 80% ACN (buffer B). The LC gradient conditions were 5% to 40% buffer B in 60 min, 40% to 100% buffer B in 5 min, and 100% buffer B in 10 min. We operated the LTQ Orbitrap XL system in the positive ion mode, and full-scan MS spectra (m/z 300–600) were acquired on the Orbitrap analyzer with a resolution of 60,000 at m/z 400. Raw files from LC-MS/MS were analyzed using MaxQuant software version v2.6.7.0.

### 4.13. TCGA Data Analysis

We downloaded 376 TCGA OV RNA-Seq level 3 read count data (serous type) from the GDC Data Portal (https://portal.gdc.cancer.gov/, accessed on 5 May 2025). We used the GENCODE version 22 gene annotation file from GDC Reference Files (https://gdc.cancer.gov/about-data/gdc-data-processing/gdc-reference-files, accessed on 5 May 2025), as utilized by the TCGA program. We searched for follow-up clinical information in the PanCanAtlas publication (https://gdc.cancer.gov/about-data/publications/pancanatlas, accessed on 5 May 2025). We calculated the TPM value for each gene in each sample based on the TPM calculation, and the formula is below. First, we investigated genes associated with the survival time of all ovarian cancer patients using an R function, COXPH (Cox Proportional-Hazards Model), with the TPM value of each gene. We calculated the hazard ratio (HR) and the Wald test of the *p*-value in each gene, and then each gene was filtered with an HR ≥ 2-fold and a *p*-value less than ≤ 0.05. For progress-free survival analysis, we calculated the best cut point TPM value for each gene by using the “cutup” function from the R package, survMisc, and dividing samples into two groups (high and low TPM groups). Finally, survival analyses were used in the R package, Survminer, and generated Kaplan–Meier plots. We calculated the best cut point value by splitting patients into high and low expression, an auto-select, and computed all possible cut point values between the lower and upper quartiles. We performed the best threshold as a cut point. We analyzed the microarray data from GEO and TCGA for all subtypes, RNA-seq data from the TCGA dataset for all subtypes, and serous types for survival analysis. We plotted the survival curve by overall survival (OS) and progression-free survival (PFS) using 1656 and 1435 patients, respectively, for GEO and 376 patients for TCGA data.

A cohort of endometrial tumors was initially created on the GDC portal (https://portal.gdc.cancer.gov/, accessed on 5 May 2025), and gene expression data and pertinent clinical data were extracted, including stage, follow-up duration, vital status, and recurrence. The final 404 samples can be obtained. The endometrial tumor samples were classified into two groups (high and low expression) using the survminer R package (version 0.4.9) with survival information and the addition of UBE2NL/HIST2H3PS2 gene expression TPM values, respectively. Ultimately, a survival analysis was conducted to examine the impact of early-stage versus advanced-stage disease, the four stages of cancer, and UBE2NL/HIST2H3PS2 high versus low gene expression groups.

### 4.14. Invasion Assay

We used Matrigel-coated transwell chambers (Corning Incorporated, New York, NY, USA) inserted into 24-well cell culture plates for invasion assays. We added SKOV3 cells, ES2 cells, or MSC-OCSPCs, (3 × 10^4^ cells in 0.1 mL of serum-free medium) to the upper chamber, and culture medium (McCoy’s 5A medium) in the lower chamber with a serum-free condition for the negative control, or containing 10% FBS for the positive control, or added culture medium (McCoy’s 5A medium) with a serum-free condition and treating EXs (18 µg) from ES2, ES2 TS, ES2TR, and ES2TR TS cell extracts. We cultivated the cells for 12 and 24 h, and cells that invaded the inserts were fixed in formalin solution for 3 min, stained with crystal violet, and counted in three random microscope fields (Olympus BX3, Olympus, Tokyo, Japan) at a magnification of 40×, 100×, or 200×.

### 4.15. In Vivo Animal Experiments

We purchased female null mice (BALB/cAnN.Cg-Foxn1nu/CrlNarl) from the National Animal Center (Taipei, Taiwan). The Institutional Animal Care and Use Committee of Cathay General Hospital approved all experiments. In experiment 1, null mice at 5–7 weeks of age (10 mice/group) were injected intraperitoneally with 1 × 10^6^ ES2 (control group) or 1 × 10^6^ ES2/shUBE2NL cells. The survival rate of each group was monitored daily and analyzed using Kaplan–Meier survival curves. A log-rank test was used to determine statistical significance, and the endpoint was defined when 80% of the control group mice had expired. In experiment 2, null mice at 5–7 weeks of age (5 mice/group) were injected intraperitoneally with 1 × 10^6^ ES2 or 1 × 10^6^ ES2/shUBE2NL cells; 1 × 10^6^ SKOV3-over luciferase or 1 × 10^6^ SKOV3/HISH3T2PS2 cells. At the endpoint, the mice were euthanized, and the severity of ascites, tumor dissemination, and peritoneal tumor growth was evaluated. The volume of ascitic fluid was measured, and disseminated tumors were counted and measured using calipers. In experiment 3, null mice at 5–7 weeks of age (5 mice/group) were injected intraperitoneally with 1 × 10^6^ SKOV3-over luciferase and 10 μg of EXs from SKOV3-over luciferase cells or 1 × 10^6^ SKOV3/HISH3T2PS2 cells and 10 μg of EXs from SKOV3/HISH3T2PS2 cells were intraperitoneally injected on day 0, and administered respective EXs twice weekly for up to six weeks to evaluate their effect on tumor progression. At the endpoint, mice underwent exploratory laparotomy for tumor dissemination analysis and careful examination of the tumor dissemination and growth in the peritoneal cavity of mice under euthanasia. Tumor burden was assessed by measuring tumor nodules, body weight changes, and ascites severity. The body weight of mice was measured, recorded, and compared with the body weight change every week. Tumor weights were recorded following euthanasia at the endpoint. We measured the number and size of metastatic tumor nodules, the severity of ascites by degrees of distension in the abdominal cavity, and the amounts of ascites in mice under euthanasia. Disseminated tumors were counted and measured in the greatest diameter using calipers.

### 4.16. Statistical Analysis

Statistical analysis was performed using SPSS 16.0. Data are expressed as a mean ± SD. Two-group comparisons used Student’s *t*-test; multiple groups used the Kruskal–Wallis test. Survival curves were generated and analyzed using Kaplan–Meier and log-rank tests. Cox regression models estimated hazard ratios. Statistical significance was defined as *p* < 0.05 (* *p* < 0.05, ** *p* < 0.01, *** *p* < 0.001).

## Figures and Tables

**Figure 1 ijms-26-04833-f001:**
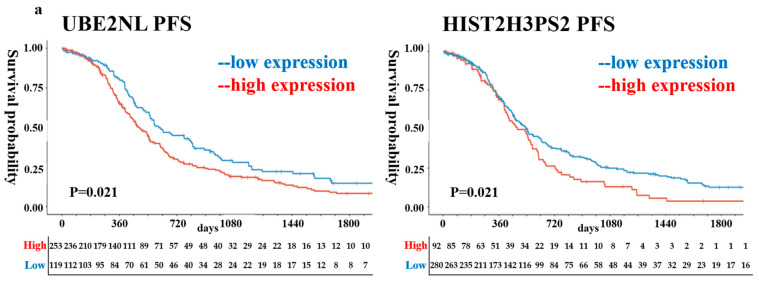
Progression-free survival (PFS) and overall survival (OS) based on UBE2NL and HIST2H3PS2 expression in epithelial ovarian cancer (EOC) patients. Kaplan–Meier survival curves using data from TCGA (**a**,**b**) show that patients with low expression of UBE2NL or HIST2H3PS2 had significantly better PFS and OS compared to those with high expression; (**c**,**d**) show that patients with low expression of UBE2NL and HIST2H3PS2 had better PFS and OS than those with high expression in both genes.

**Figure 2 ijms-26-04833-f002:**
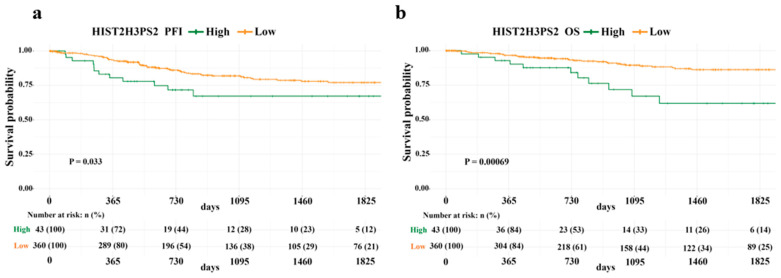
Progression-free survival (PFS) and overall survival (OS) based on HIST2H3PS2 expression in endometrial cancer (EC) patients. Kaplan–Meier survival analysis using TCGA data shows that EC patients with low HIST2H3PS2 expression had significantly better PFS (**a**) and OS (**b**) across all clinical stages than those with high HIST2H3PS2 expression.

**Figure 3 ijms-26-04833-f003:**
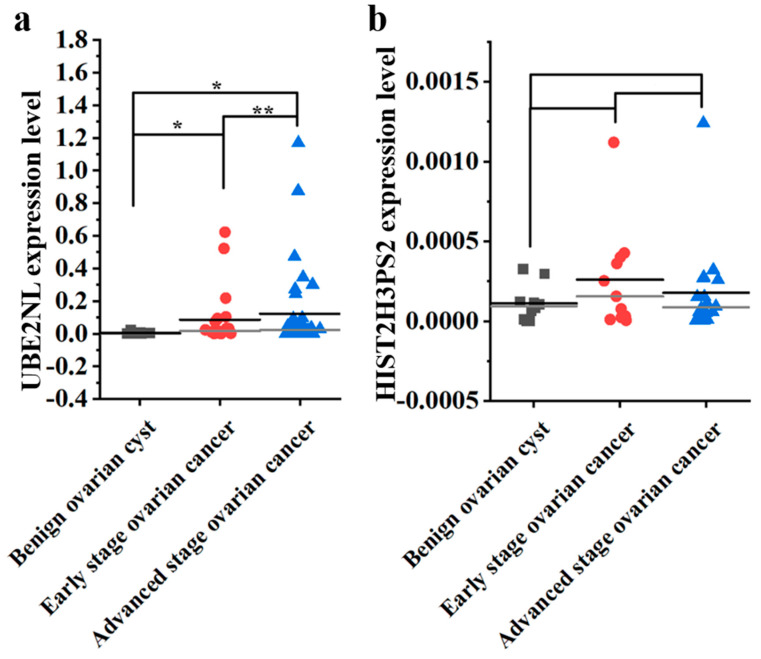
Distribution of UBE2NL and HIST2H3PS2 mRNA in ovarian tissues. The dot plot shows mRNA expression levels of (**a**) UBE2NL and (**b**) HIST2H3PS2 in benign ovarian cysts (n = 10), early-stage (n = 21), and advanced-stage ovarian cancer (n = 35). Mean ± SD expressed in values. The Student’s *t*-test determined the significance. * represents *p* < 0.05; ** represents *p* < 0.01.

**Figure 4 ijms-26-04833-f004:**
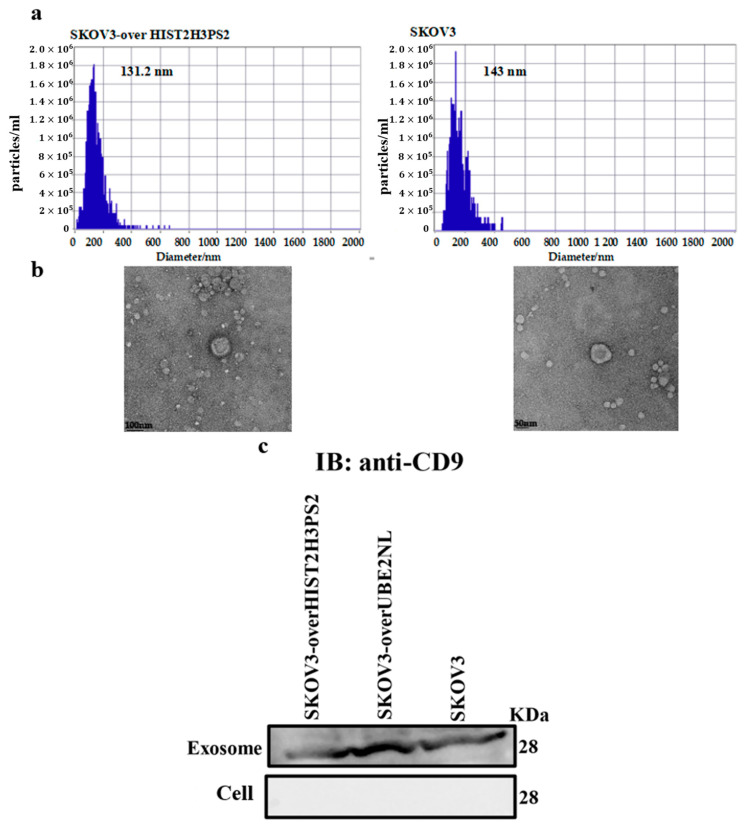
Characterization of exosomes derived from SKOV3 and SKOV3-overexpressing HIST2H3PS2 cells. (**a**) Nanoparticle tracking analysis (NTA) showing the particle size distribution and concentration of exosomes derived from SKOV3 and SKOV3/HIST2H3PS2 cells. The *x*-axis represents particle size (nm), and the *y*-axis shows particle concentration (×10^6^/mL). (**b**) Transmission electron microscopy (TEM) images of the typical morphology of exosomes from SKOV3 and SKOV3/HIST2H3PS2 cells were displayed. (**c**) Western blot analysis for the exosome marker CD9. CD9 was present in exosomes derived from SKOV3, SKOV3/UBE2NL, and SKOV3/HIST2H3PS2 cells but not in the corresponding cell lysates, confirming exosome specificity.

**Figure 5 ijms-26-04833-f005:**
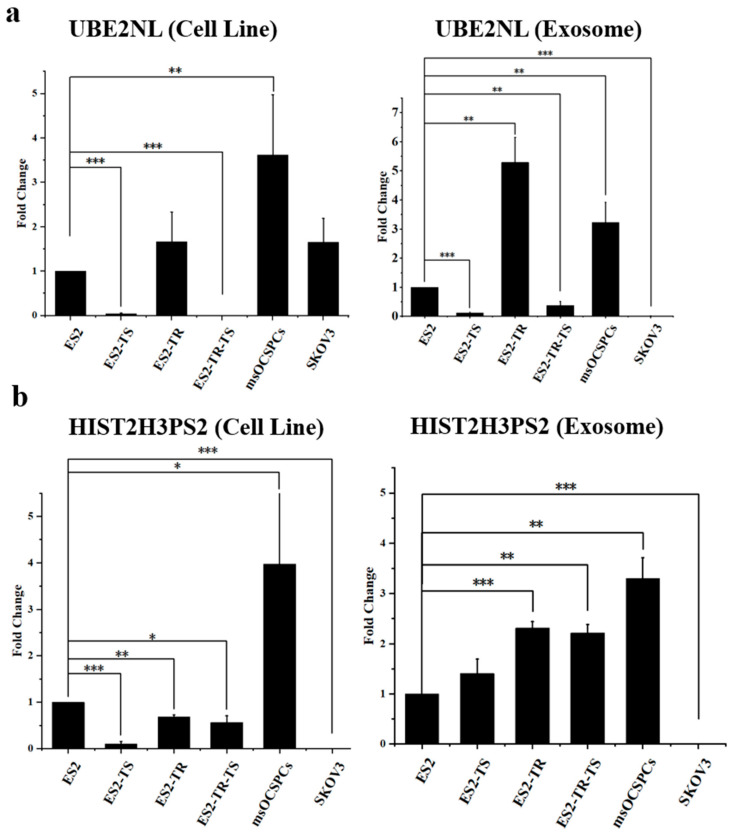
Differential expressions of UBE2NL and HIST2H3PS2 in EOC cells and their exosomes. UBE2NL (**a**) and HIST2H3PS2 (**b**) were detected in exosomes and cell lysates of more aggressive EOC cell lines, including ES2, ES2TR, spheres derived from ES2 cells (ES2TS), spheres derived from ES2TR cells (ES2TRTS), SKOV3, and msOCSPCs (MSC-OCSPCs), but were absent in exosomes derived from the less aggressive SKOV3 cells. * represents *p* < 0.05; ** represents *p* < 0.01; *** represents *p* < 0.001.

**Figure 6 ijms-26-04833-f006:**
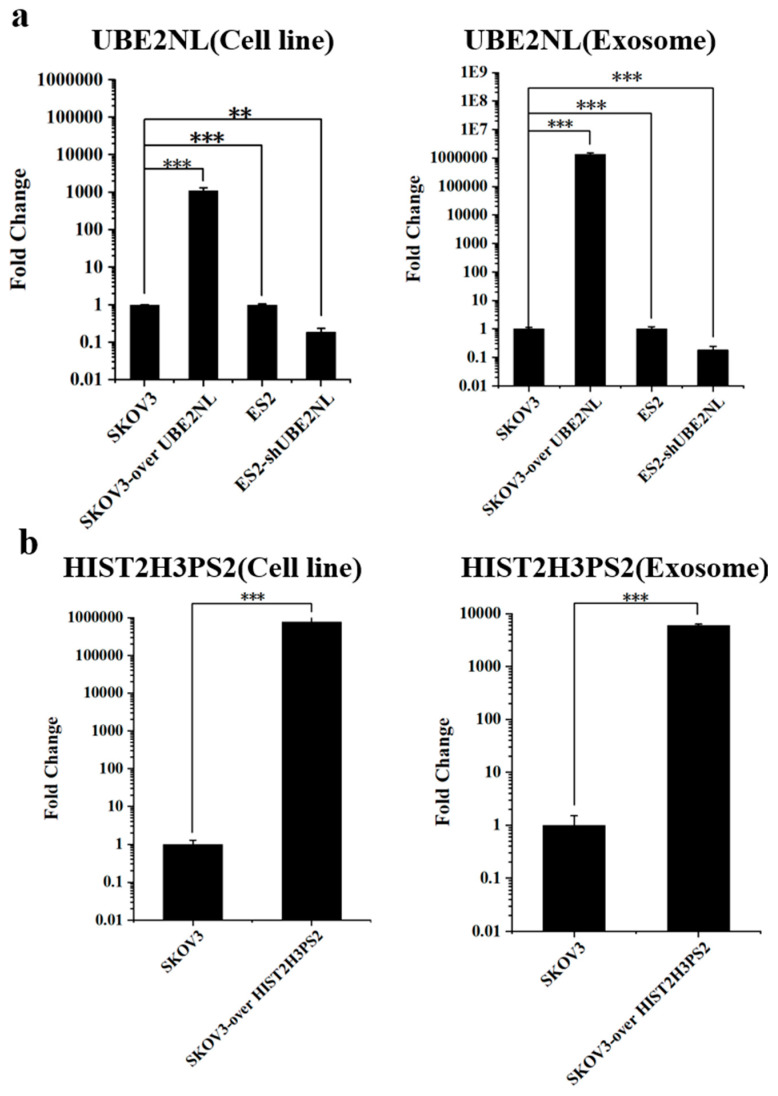
Expression of UBE2NL and HIST2H3PS2 in EOC cell lines and their derived exosomes. (**a**) Quantitative RT-PCR analysis showed that UBE2NL expression was higher in SKOV3/UBE2NL cells and their exosomes compared to SKOV3 controls; on the contrary, it was significantly reduced in ES2/shUBE2NL cells and their exosomes compared to wild-type ES2. (**b**) HIST2H3PS2 expression was also elevated in SKOV3/HIST2H3PS2 cells and their exosomes relative to SKOV3. ** represents *p* < 0.01; *** represents *p* < 0.001.

**Figure 7 ijms-26-04833-f007:**
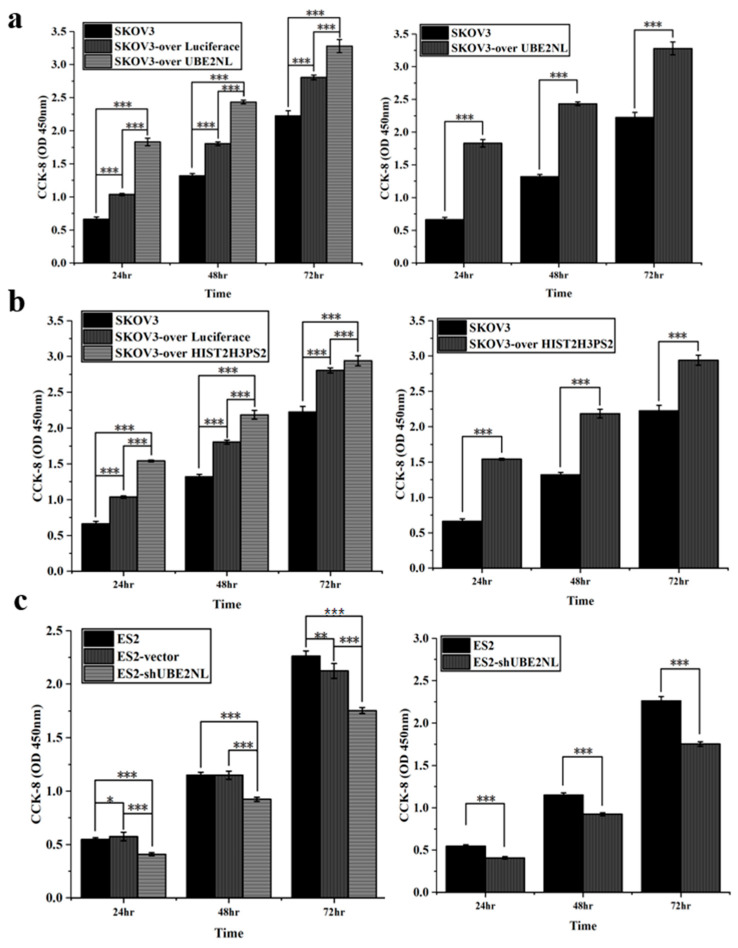
Cell viability analysis of EOC cells using CCK-8 assay. (**a**) SKOV3/UBE2NL cells exhibited significantly higher viability compared to wild-type SKOV3 cells. (**b**) Similarly, SKOV3/HIST2H3PS2 cells showed significantly increased viability relative to SKOV3 controls. (**c**) In contrast, ES2/shUBE2NL cells demonstrated significantly reduced viability compared to wild-type ES2 cells. * represents *p* < 0.05; ** represents *p* < 0.01; *** represents *p* < 0.001.

**Figure 8 ijms-26-04833-f008:**
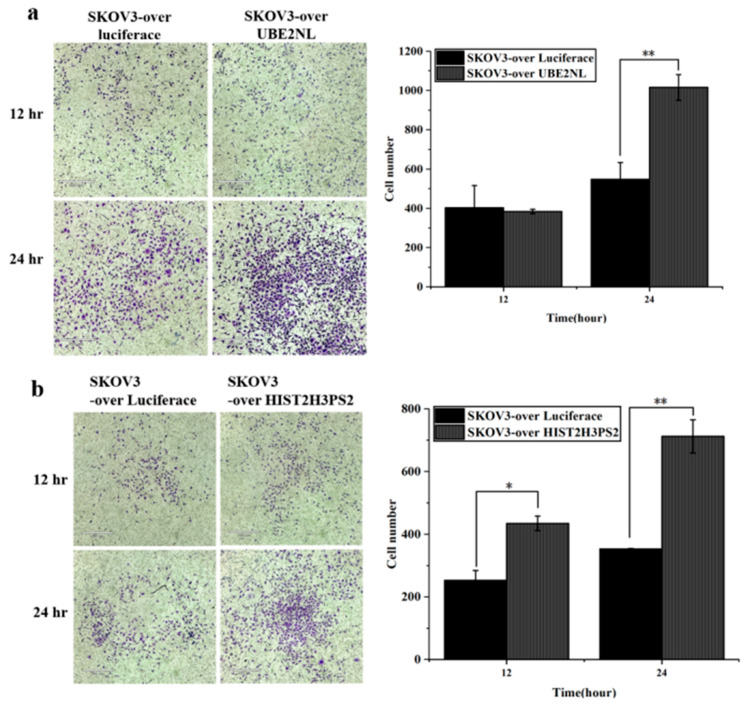
Invasion ability of SKOV3 cells overexpressing luciferase, UBE2NL, or HIST2H3PS2. Transwell invasion assays revealed that SKOV3 cells overexpressing (**a**) UBE2NL or (**b**) HIST2H3PS2 exhibited significantly greater invasive ability than SKOV3 cells overexpressing luciferase (control) at a microscope’s magnification of 100×. * represents *p* < 0.05; ** represents *p* < 0.01.

**Figure 9 ijms-26-04833-f009:**
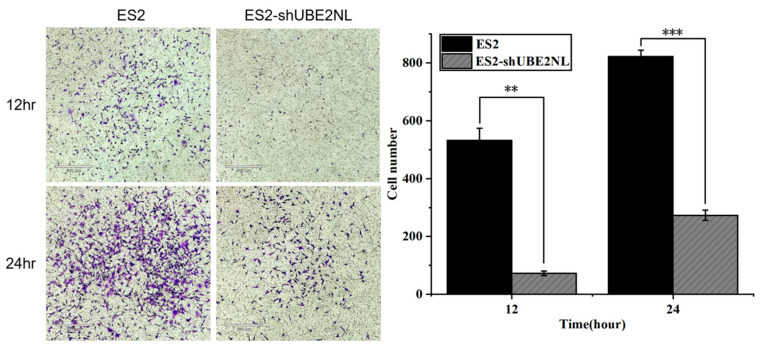
Invasion ability of ES2 and ES2/shUBE2NL cells. ES2/shUBE2NL cells exhibited significantly reduced invasion ability compared to wild-type ES2 cells, indicating that UBE2NL contributes to the invasive phenotype in EOC cells at a microscope’s magnification of 100×. ** represents *p* < 0.01; *** represents *p* < 0.001.

**Figure 10 ijms-26-04833-f010:**
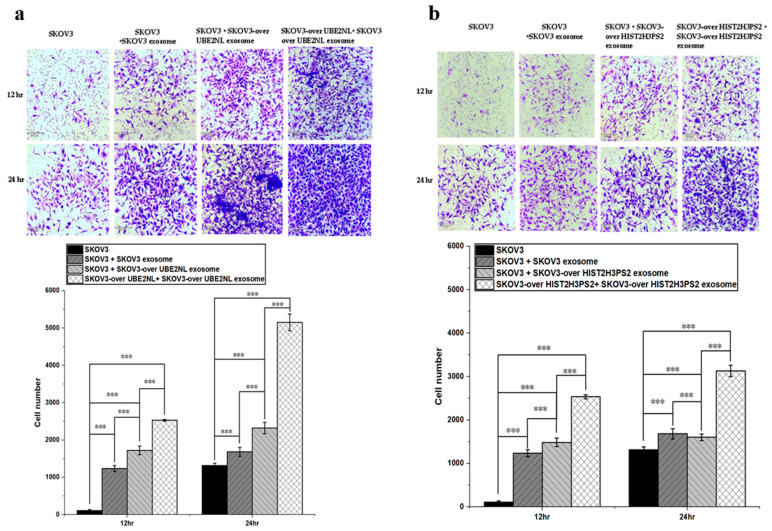
Invasion ability of SKOV3 cells treated with UBE2NL- or HIST2H3PS2-enriched exosomes. (**a**) SKOV3 cells treated with SKOV3 EXs, SKOV3/UBE2NL EXs, or co-cultured with SKOV3/UBE2NL cells and their EXs showed significantly greater invasion ability than untreated SKOV3 cells at a microscope’s magnification of 100×. (**b**) Similarly, SKOV3 cells treated with SKOV3 EXs, SKOV3/HIST2H3PS2 EXs, or co-cultured with SKOV3/HIST2H3PS2 cells and their EXs exhibited significantly enhanced invasion compared to untreated SKOV3 cells at a microscope’s magnification of 100×. *** represents *p* < 0.001.

**Figure 11 ijms-26-04833-f011:**
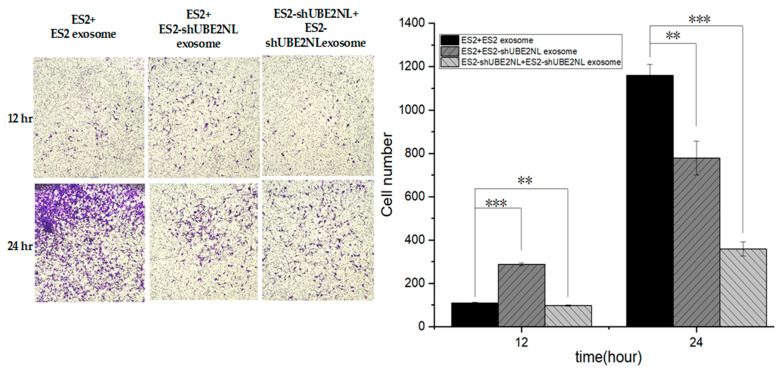
Invasion ability of ES2 cells treated with ES2 or ES2/shUBE2NL-derived exosomes. ES2 cells and ES2/shUBE2NL cells treated with EXs from ES2/shUBE2NL showed significantly decreased invasion compared to ES2 cells treated with EXs from wild-type ES2 cells at a microscope’s magnification of 100×. ** represents *p* < 0.01; *** represents *p* < 0.001.

**Figure 12 ijms-26-04833-f012:**
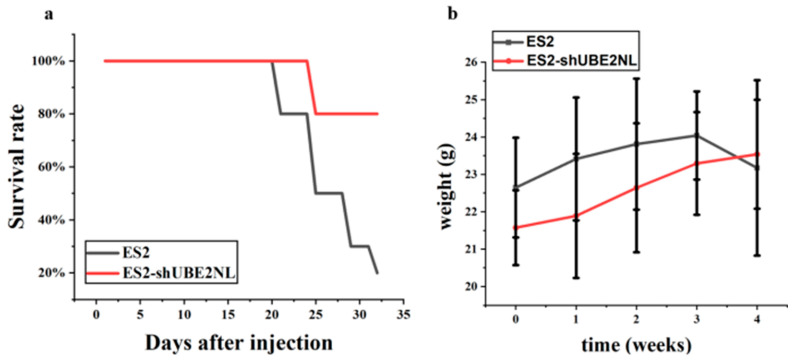
In vivo survival analysis of mice injected intraperitoneally with ES2 or ES2/shUBE2NL cells. (**a**) Kaplan–Meier survival curves show significantly improved survival in the ES2/shUBE2NL group compared to the ES2 group (*p* < 0.05, log-rank test). (**b**) No significant differences were observed in body weight before and after cell injection.

**Figure 13 ijms-26-04833-f013:**
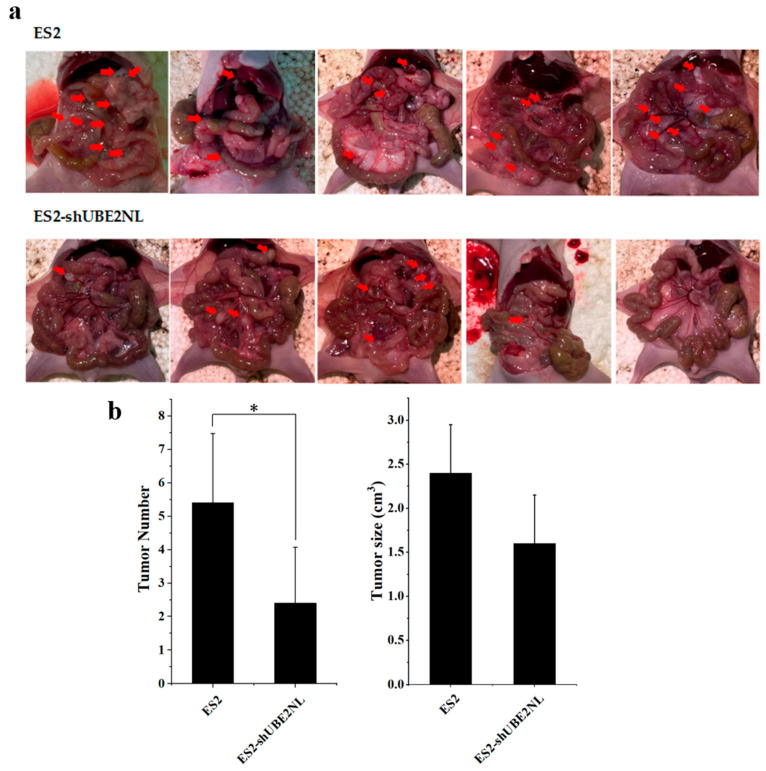
Disseminated tumors in the peritoneal cavity of mice. (**a**) Representative images and quantification of disseminated tumors in mice intraperitoneally injected with 1 × 10^6^ cells. Red arrows indicate the location of tumor nodules. (**b**) Mice injected with ES2/shUBE2NL cells exhibited significantly fewer tumor nodules than those injected with ES2 cells. * represents *p* < 0.05.

**Figure 14 ijms-26-04833-f014:**
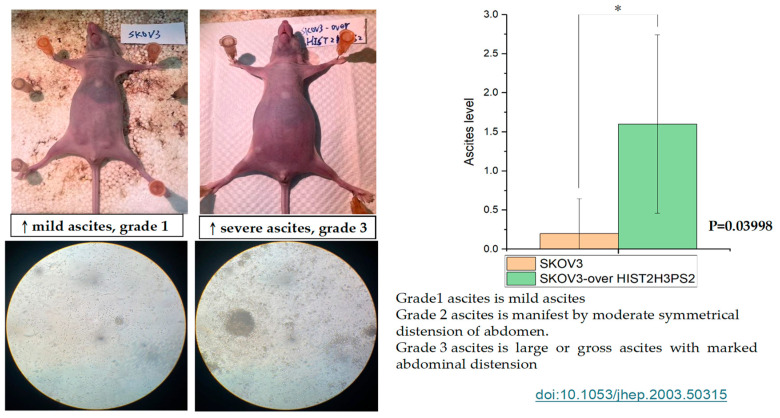
Severity of ascites formation in mice injected with SKOV3 or SKOV3/HIST2H3PS2 cells. Mice injected with SKOV3/HIST2H3PS2 cells developed significantly more severe ascites than those injected with SKOV3 cells. The bar chart shows the severity of ascites observed grossly and histologically at a microscope’s magnification of 40× in the peritoneal cavity of mice following intraperitoneal injection of 1 × 10^6^ SKOV3 or SKOV3/HIST2H3PS2 cells [30]. * represents *p* < 0.05.

**Figure 15 ijms-26-04833-f015:**
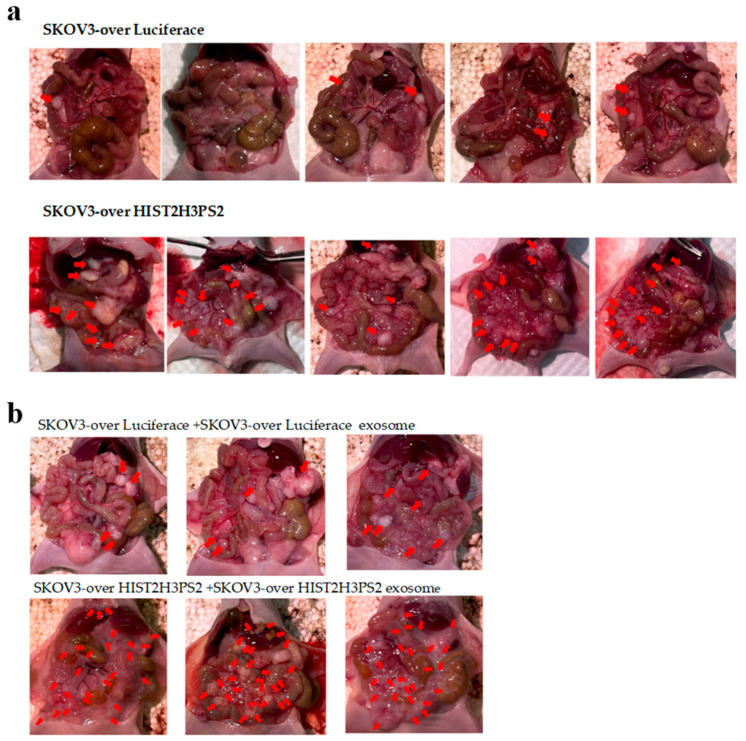
Visual and quantitative assessment of disseminated tumors in the peritoneal cavity of mice. (**a**) Representative photographs of peritoneal tumor dissemination in mice injected intraperitoneally with SKOV3-overexpressing luciferase (SKOV3/Luc) cells or SKOV3/HIST2H3PS2 cells. The average number of disseminated tumors was significantly lower in mice injected with SKOV3/Luc cells than in those injected with SKOV3/HIST2H3PS2 cells. Red arrows indicate tumor nodules. (**b**) Quantification of tumor number and size. Mice injected with 1 × 10^6^ SKOV3/HIST2H3PS2 cells plus 10 μg EXs showed a significantly greater number of disseminated tumor nodules than those injected with 1 × 10^6^ SKOV3 cells plus 10 μg EXs. In contrast, the average tumor size was smaller in the SKOV3/HIST2H3PS2 + EX group, suggesting a more widespread but smaller tumor burden.

**Figure 16 ijms-26-04833-f016:**
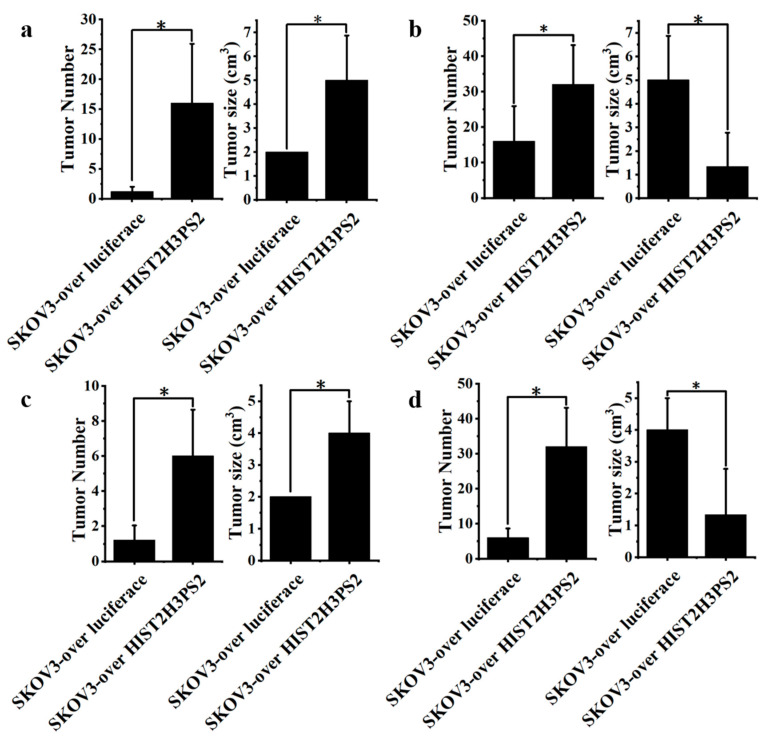
Quantification of disseminated tumor number and size in the peritoneal cavity of mice. (**a**) Mice injected with SKOV3-overexpressing luciferase (SKOV3/Luc) cells had significantly fewer peritoneal tumors than those injected with SKOV3/HIST2H3PS2 cells. (**b**) Mice injected with 1 × 10^6^ SKOV3/HIST2H3PS2 cells plus 10 μg EXs showed a significantly higher number of disseminated tumors than those injected with SKOV3/HIST2H3PS2 cells alone. (**c**) Similarly, mice receiving SKOV3 cells plus 10 μg EXs exhibited significantly more disseminated tumors than those injected with SKOV3 cells alone. (**d**) Tumor number was significantly greater in mice injected with SKOV3/HIST2H3PS2 cells plus 10 μg EXs than in mice receiving SKOV3 cells plus 10 μg EXs. In contrast, the average tumor size was significantly smaller in the SKOV3/HIST2H3PS2 + EX group compared to the SKOV3 + EX group, suggesting more widespread but smaller tumor dissemination. * represents *p* < 0.05.

**Figure 17 ijms-26-04833-f017:**
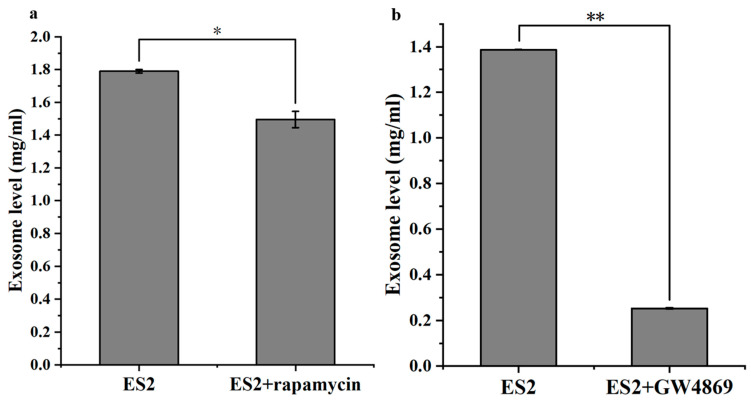
Exosome secretion levels in ES2 cells treated with exosome inhibitors. Exosome concentrations (mg/mL) were measured from the culture supernatants of 1 × 10^6^ ES2 cells treated with (**a**) 500 nM rapamycin or (**b**) 10 μM GW4869. Both treatments significantly reduced the level of secreted exosomes compared to untreated controls. * represents *p* < 0.05; ** represents *p* < 0.01.

## Data Availability

Data is contained within the article and Appendix A.

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
