# Peer review of "Extracellular-Vesicle-Associated UBE2NL and HIST2H3PS2 Promote Tumor Aggressiveness and Metastasis in Gynecologic Cancer"

_ijms, 2025, doi:10.3390/ijms26104833_

Round 1
Reviewer 1 Report
Comments and Suggestions for Authors
In this manuscript, the authors identify two novel genes, UBE2NL and HIST2H3PS2, and demostrate their significant roles in promoting cell growth and invasiveness in epithelial ovarian cancer (EOC) cells. A key finding is the mechanism by which EOC cells utilize extracellular vesicles (EVs) to transfer these genes to neighboring cells, thereby promoting tumorigenesis. The manuscript presents a complete and well-supported narrative based on a substantial amount of data.
However, the manuscript requires several revisions before it can be considered for publication. Specific concerns include:
- Survival Analysis (Figure 1): The authors demonstrate a negative association between individual UBE2NL and HIST2H3PS2 expression and patient survival. An analysis of the combined prognostic value of these two genes would be of interest.
- Data Visualization (Table 1): Consider using dot plots to better illustrate the distribution of sample data.
- EV Production (Lines 175-176): The claim that HIST2H3PS2-overexpressed SKOV3 cells produce more EVs requires stronger supporting data. The observed increase could potentially be attributed to the known proliferative effects of HIST2H3PS2 leading to a higher cell number.
- Western Blotting (Figure 3): The Western blot data contains duplicate entries. Additionally, the label "protein" should be corrected to "cell" when comparing exosomal and cellular protein expression for CD9.
- COL6A3 Data (Section 2.4 & Figure 4): The relevance of the COL6A3-related data is unclear, as the manuscript does not discuss its role in regulating UBE2NL and HIST2H3PS2 Please clarify the significance of this data. Additionally, define the abbreviation "TR160-TS" in Figure 4.
- UBE2NL Expression (Lines 210-231 & Figure 4): The seemingly contradictory finding of high UBE2NL expression in non-aggressive SKOV3 cells compared to aggressive ES2 cells requires explanation before the interpretation provided.
- Axis Labels (Figures 12 & 14): Please add units to the Y-axes in Figure 12 and Figure 14.
- Exosome Nomenclature: Revise instances of "HIST2H3PS2- or UBE2NL-derived exosomes" to "HIST2H3PS2- or UBE2NL-enriched exosomes" throughout the manuscript.
- Exosome Isolation method (Section 4.4): The exosome isolation protocol requires more detail, including cell culture conditions and whether differential centrifugation was employed to remove larger EVs.
- Manuscript Quality: The manuscript requires thorough editing and proofreading to address lexical and mechanical errors and improve overall flow. The figures should be reorganized for clarity and enhanced reader-friendliness.

The manuscript requires thorough editing and proofreading to address lexical and mechanical errors and improve overall flow. The figures should be reorganized for clarity and enhanced reader-friendliness.
Author Response
In this manuscript, the authors identify two novel genes, UBE2NL and HIST2H3PS2, and demostrate their significant roles in promoting cell growth and invasiveness in epithelial ovarian cancer (EOC) cells. A key finding is the mechanism by which EOC cells utilize extracellular vesicles (EVs) to transfer these genes to neighboring cells, thereby promoting tumorigenesis. The manuscript presents a complete and well-supported narrative based on a substantial amount of data.
However, the manuscript requires several revisions before it can be considered for publication. Specific concerns include:
- Survival Analysis (Figure 1): The authors demonstrate a negative association between individual UBE2NL and HIST2H3PS2 expression and patient survival. An analysis of the combined prognostic value of these two genes would be of interest.
Response. Thank you for your comment. We remade Figure 1 and added new Figures 1, c, and d to analyze the combined prognostic value of these two genes. In lines 122-125, we add the sentences “ When both genes of UBE2NL and HIST2H3PS2 were considered, progression-free survival and overall survival were better in patients with low expressions of both genes than in those with high expressions (p=0.017 for PFS, p=0.022 for OS) (Figure 1, c and d). We revised the sentences in Figure 1 legend “ Kaplan–Meier survival curves using data from TCGA (a and b) show that patients with low expression of UBE2NL or HIST2H3PS2 had significantly better PFS and OS compared to those with high expression; (c and d) show that patients with low expression of UBE2NL and HIST2H3PS2 were better PFS and OS than those with high expression in both genes.
After an internal review, we decided to remove the original Figure 1C chart from the revised manuscript because some of the specimens used in this section were collected at an early stage without formal IRB approval at the time, and they linked to long-term survival data which might not meet the unidentifiable standard under current ethical standards. We remove the image to ensure that the study complies with human subjects’ research laws and ethical review regulations and maintains the highest research integrity.
- Data Visualization (Table 1): Consider using dot plots to better illustrate the distribution of sample data.
Response. Thank you for your comment. We used dot plots to illustrate the distribution of ovarian tissues in Figure 3. We reviewed the original data and recalculated the mean value in each group for statistical significance. We corrected the data and rewrote the paragraph in lines 159-168. The mean mRNA expression levels of UBE2NL showed a significant increase across disease stages. Specifically, UBE2NL expression was significantly higher in advanced-stage ovarian cancer tissues compared to early-stage cancer and benign ovarian cysts (mean ± SD: 0.1225 ± 0.2508 vs. 0.0862 ± 0.1662 vs. 0.0037 ± 0.0069; p < 0.001, by Kruskal-Wallis test).
For HIST2H3PS2, the mean expression level was higher in advanced-stage and early-stage ovarian cancer compared to benign cysts (mean ± SD: 0.0002 ± 0.0003 vs. 0.0003 ± 0.0003 vs. 0.0001 ± 0.0001; p >0.05, by Kruskal-Wallis test). The dot plots illustrate the distribution of UBE2NL and HIST2H3PS2 mRNA expression levels in ovarian tissues among benign ovarian cysts, early-stage ovarian cancer, and advanced-stage ovarian cancers in Figure 3.
We revised the legend in Figure 3 as follows. Figure 3. Distribution of UBE2NL and HIST2H3PS2 mRNA in ovarian tissues. The dot plot shows mRNA expression levels of (a) UBE2NL and (b) HIST2H3PS2 in benign ovarian cysts (n=10), early-stage (n=21), and advanced-stage ovarian cancer (n=35). Mean ± SD expressed in values. The student’s t-test decided the significance.* represents p<0.05; ** represents<0.01.
- EV Production (Lines 175-176): The claim that HIST2H3PS2-overexpressed SKOV3 cells produce more EVs requires stronger supporting data. The observed increase could potentially be attributed to the known proliferative effects of HIST2H3PS2 leading to a higher cell number.
Response. Thank you for your comment. We deleted the sentences “suggesting that SKOV3/HIST2H3PS2 cells produce more abundant EXs”
- Western Blotting (Figure 3): The Western blot data contains duplicate entries. Additionally, the label "protein" should be corrected to "cell" when comparing exosomal and cellular protein expression for CD9.
Response. Thank you for your comment. We remade the new Figure 4 and the original Figure 3 and deleted the uncropped gel picture, which was presented as supplemental data.
- COL6A3 Data (Section 2.4 & Figure 4): The relevance of the COL6A3-related data is unclear, as the manuscript does not discuss its role in regulating UBE2NL and HIST2H3PS2 Please clarify the significance of this data. Additionally, define the abbreviation "TR160-TS" in Figure 4.
Thank you for your comment. We remade the new Figure 5 and the original Figure 4 and deleted COL6A3-related data. Additionally, we used ES2TR-TS to replace TR160-TS, which represented spheres derived from ES2TR cells.
- UBE2NL Expression (Lines 210-231 & Figure 4): The seemingly contradictory finding of high UBE2NL expression in non-aggressive SKOV3 cells compared to aggressive ES2 cells requires explanation before the interpretation provided.
Response. We sincerely thank the reviewer for this valuable and insightful comment.
As noted, although ES2 cells are generally recognized as more aggressive than SKOV3 cells, the basal mRNA expression level of UBE2NL was lower in ES2 cells compared to SKOV3 cells. We believe this discrepancy may arise from intrinsic molecular differences between the two cell lines, as they are derived from distinct histological subtypes of ovarian cancer. Moreover, the basal expression levels may not fully reflect the functional impact of UBE2NL on tumor aggressiveness. In our subsequent functional assays, we observed that overexpression of UBE2NL markedly enhanced aggressive behaviors in EOC cells, supporting its role in promoting malignancy beyond basal expression levels alone.
We have added a clarification in Section 2.4 of the revised manuscript. “Interestingly, although ES2 cells are generally considered more aggressive than SKOV3 cells, the basal mRNA expression level of UBE2NL was lower in ES2 cells compared to SKOV3 cells. This discrepancy may be due to intrinsic molecular differences between the two cell lines from distinct histological subtypes of ovarian cancer. Moreover, the basal expression levels may not fully reflect the functional impact of UBE2NL on tumor aggressiveness. In our subsequent functional assays, we observed that overexpression or exosome-mediated transfer of UBE2NL markedly enhanced aggressive behaviors in EOC cells, supporting its role in promoting malignancy beyond basal expression levels alone.
Axis Labels (Figures 12 & 14): Please add units to the Y-axes in Figure 12 and Figure 14.
Response. Thank you for your comment. We remade the new Figure 13 and Figure 16 and the original Figures 12 and 14 and added units to the Y-axis.
- Exosome Nomenclature: Revise instances of "HIST2H3PS2- or UBE2NL-derived exosomes" to "HIST2H3PS2- or UBE2NL-enriched exosomes" throughout the manuscript.
Response. Thank you for your comment. We revised instances of "HIST2H3PS2- or UBE2NL-derived exosomes" to "HIST2H3PS2- or UBE2NL-enriched exosomes" throughout the manuscript.
- Exosome Isolation (Section 4.4): The exosome isolation protocol requires more detail, including cell culture conditions and whether differential centrifugation was employed to remove larger EVs.
Response. Thank you for your valuable comment. We have revised Section 4.4 to include more comprehensive details on the exosome isolation procedure, including specific cell culture conditions and centrifugation steps to remove larger extracellular vesicles (EVs). The revised section is as follows: 4.4. Exosome Isolation “To collect exosomes, we seeded ES2, ES2TR, and SKOV3 cells at a density of 2 × 10⁶ cells per 10 cm dish in 10 mL of complete culture medium. Once the cells reached approximately 95% confluence, the medium was replaced with serum-free McCoy’s 5A medium and incubated for 24 hours. Conditioned media were harvested and centrifuged at 3000 × g for 15 minutes at 4 °C to remove cells and cellular debris, eliminating larger EVs and apoptotic bodies. The clarified supernatants were transferred to a fresh tube and mixed with ExoQuick-TC™ solution (System Biosciences, USA) at a 1:5 (v/v) ratio according to the manufacturer’s instructions. The mixture was gently inverted to mix and incubated overnight at 4 °C. Exosomes were subsequently pelleted by centrifugation at 1500 × g for 30 minutes, followed by an additional centrifugation at 1500 × g for 5 minutes to remove residual ExoQuick-TC reagent. The final exosome pellet was gently resuspended in 100 μL of sterile 1× PBS and stored at −80 °C until further use.
- Manuscript Quality: The manuscript requires thorough editing and proofreading to address lexical and mechanical errors and improve overall flow. The figures should be reorganized for clarity and enhanced reader-friendliness
Response. Thank you for your comment. We revised the manuscript through editing and proofreading to address lexical and mechanical errors and improve the overall flow. The figures have been remade and reorganized for clarity and enhanced reader-friendliness.

Reviewer 2 Report
Comments and Suggestions for Authors
The authors did very important and interesting research on extracellular vesicles using cell culture and in vivo models. I personally like their approaches, the way they planned their studies. I have some suggestions to improve the manuscript.
Please mention all details in the materials and methods section. For example, how the cells were seeded, what the cell number per mL was, and what the total volume of cell culture media used for the EX isolation, etc. (all details).
What was the total number of EX when the EX lysate was prepared for western blot? Why CD-9 was used, not CD63 or CD81 etc.
Figure-3 top: Y-axis "ml/particles" should be "particles/ml".
All other graphs Y-axis "Fold" could be written as "Fold change" to make better sense.
All figures: please use a, b, c, d, etc to clearly label them to make them easy to describe in the text.
EX concentration was measured using Bradford protein assay. It seems the EX protein concentration in the EX-lysate was measured, right? Please write clearly.
Please discuss along with some published related research articles and cite them. Discuss the similarities or dissimilarities with them, and what could be the future target based on the results.
I would recommend uploading the TCGA data analysis pipelines/details as a supplementary file. That would be helpful for the readers and future researchers as well.
Author Response
The authors did very important and interesting research on extracellular vesicles using cell culture and in vivo models. I personally like their approaches, the way they planned their studies. I have some suggestions to improve the manuscript.
Please mention all details in the materials and methods section. For example, how the cells were seeded, what the cell number per mL was, and what the total volume of cell culture media used for the EX isolation, etc. (all details).
Response:
Thank you for your valuable comment. We revised the Materials and Methods section in Sections 4.2 and 4.4. We provided comprehensive details of the exosome isolation procedures to address how the cells were seeded, what the cell number per mL was seeded, and what the total volume of cell culture media used. The revised section is as follows:
4.2. Cell Lines and Culture
ES2 and SKOV3 cell lines were from the American Type Culture Collection (ATCC). ES2 is a cell line exhibiting fibroblast-like morphology with clear cell carcinoma features. SKOV3 is a cell line displaying epithelial morphology with gynecologic high-grade carcinoma characteristics derived from uterine or ovarian serous carcinoma. Cells cultured in McCoy's 5A medium supplemented with 10% FBS and maintained at 37 °C in a humidified atmosphere with 5% CO₂. Cells were seeded at a density of 2 × 10⁵ cells per 10 cm dish in 10 mL of complete medium and passaged every 2–3 days upon reaching 80–90% confluence. Paclitaxel-resistant ES2 cells (ES2TR) were developed by continuous exposure to increasing concentrations of paclitaxel, with a final selection concentration of 160 nM.
4.4. Exosome Isolation
To collect exosomes, we seeded ES2, ES2TR, and SKOV3 cells at a density of 2 × 10⁶ cells per 10 cm dish in 10 mL of complete culture medium. Once the cells reached approximately 95% confluence, the medium was replaced with serum-free McCoy’s 5A medium and incubated for 24 hours. Conditioned media were harvested and centrifuged at 3000 × g for 15 minutes at 4 °C to remove cells and cellular debris, eliminating larger EVs and apoptotic bodies. The clarified supernatants were transferred to a fresh tube and mixed with ExoQuick-TC™ solution (System Biosciences, USA) at a 1:5 (v/v) ratio according to the manufacturer's instructions. The mixture was gently inverted to mix and incubated overnight at 4 °C. Exosomes were subsequently pelleted by centrifugation at 1500 × g for 30 minutes, followed by an additional centrifugation at 1500 × g for 5 minutes to remove residual ExoQuick-TC reagent. The final exosome pellet was gently resuspended in 100 μL of sterile 1× PBS and stored at −80 °C until further use.
What was the total number of EX when the EX lysate was prepared for western blot? Why CD-9 was used, not CD63 or CD81 etc.
Response:
Thank you for the valuable comment. We added the sentences in 4.7. Western Blot Analysis, lines 562-564 Approximately 30 µg of exosomal protein (quantified by Bradford assay), corresponding to approximately 8 × 10⁹ exosome particles, as quantified by nanoparticle tracking analysis (NTA) loaded per lane for Western blot analysis of exosomes.
That yields a particle-to-protein ratio of 2.67 × 10⁸ particles/µg protein, which falls within the range reported in the literature. For example, Bok et al. (2024) reported that exosomes isolated by ultracentrifugation combined with size exclusion chromatography exhibited a particle-to-protein ratio of approximately 2.83 × 10⁸ particles/µg, supporting the consistency and purity of our exosome preparations.
We also acknowledge that this ratio can vary depending on the cell type, isolation method, and analytical settings, and we have taken these variables in our experimental design.
We chose CD9 as the exosomal marker because our preliminary analyses showed that CD9 was more abundant and consistent in our ovarian cancer-derived exosome preparations than CD63 (ref 29).
Reference:
Bok EY, Seo SY, Lee HG, Wimalasena SHMP, Kim E, Cho A, Jung YH, Hur TY, So KM, Lee SL, Do YJ.J Exosomes isolation from bovine serum: qualitative and quantitative comparison between ultracentrifugation, combination ultracentrifugation and size exclusion chromatography, and exoEasy methods. Anim Sci Technol. 2024 Sep;66(5):1021-1033. doi: 10.5187/jast.2024.e45.
[29] Ho, C.M.; Yen, T.L.; Chang, T.H.; Huang, S.H. COL6A3 Exosomes Promote Tumor Dissemination and Metastasis in Epithelial Ovarian Cancer. Int. J. Mol. Sci. 2024, 25, 8121. https://doi.org/10.3390/ijms25158121.
Figure-3 top: Y-axis "ml/particles" should be "particles/ml".
Response. Thank you for your comment. We remade the new Figure 4a in the original Figure 3a. The Y-axis has changed to particles/ml.
All other graphs Y-axis "Fold" could be written as "Fold change" to make better sense.
Thank you for your comment. We remade the new Figures 5 and 6 in the original Figures 4 and 5 to use fold changes in the y-axis.
All figures: please use a, b, c, d, etc to clearly label them to make them easy to describe in the text.
Response. Thank you for your comment. We used a, b, c, and d to label all the Figures and figure legends.
EX concentration was measured using Bradford protein assay. It seems the EX protein concentration in the EX-lysate was measured, right? Please write clearly.
Response. Thank you for your comment. We revised section 4.8. Exosome Quantification is as follows. The protein concentration of exosome lysates was determined using a Bradford protein assay (SIGMA, B6916). Briefly, the exosome sample was lysed, diluted in sterile 1X PBS, and mixed with Bradford reagent according to the manufacturer’s instructions. After incubation at room temperature, we measured the absorbance at 595 nm. We prepared the standard curve using bovine serum albumin (BSA), and the protein concentration of exosome lysates was calculated and expressed as mg/mL.
Please discuss along with some published related research articles and cite them. Discuss the similarities or dissimilarities with them, and what could be the future target based on the results.
Thank you for your comment. We revised the discussion section as follows. Our findings are consistent with previous studies demonstrating that Ubiquitin-conjugating enzymes such as UBE2N are increasingly recognized for their roles in DNA repair, immune signaling, and oncogenesis [30,31]. UBE2N, in complex with UBE2V2, modulates DNA repair proteins like PCNA and p53 through K63-linked ubiquitination, which is crucial for maintaining genomic stability [32,33]. Overexpression of UBE2N has been associated with poor prognosis, chemotherapy resistance, and enhanced NF-κB signaling in various cancers [34,35]. Although UBE2NL is less studied, its structural similarity to UBE2N and ability to dimerize with E2V2 suggest a potential role in similar pathways, particularly those regulating protein degradation and cell cycle progression [36,37].
Similarly, our observation is that HIST2H3PS2, a histone H3 pseudogene, may contribute to cancer progression through its involvement in chromatin remodeling and epigenetic regulation [38]. Epigenetic alterations such as histone acetylation and methylation are involved in ovarian cancer progression and chemotherapy drug resistance through mechanisms involving histone deacetylases (HDACs) and dysregulated oncogene or tumor suppressor expression [39,40]. Histone deacetylase (HDAC) inhibitors have shown efficacy in reversing chemotherapy resistance in ovarian cancer models, underscoring the importance of epigenetic regulators as therapeutic targets [40].
Our research presents several novel insights compared to previous studies. While the oncogenic roles of UBE2N and tumor-derived EVs have been previously reported [1,11], our work is the first to demonstrate a functional role for UBE2NL and HIST2H3PS2, particularly in the context of EV-mediated tumor progression in gynecologic cancers. Furthermore, the role of histone pseudogenes in exosome biology remains unexplored, highlighting the novelty of our findings.
Conclusions and Future Perspectives
In conclusion, this study identifies UBE2NL and HIST2H3PS2 as novel contributors to tumor aggressiveness and metastasis in epithelial ovarian and endometrial cancer. Both genes promote invasive behavior in vitro and facilitate tumor dissemination in vivo, effects enhanced by their respective exosomes. Our findings highlight the significance of EV-mediated intercellular communication in gynecologic cancer progression.
Future investigations should focus on several key areas. First, mechanistic studies are needed to elucidate the molecular signaling pathways downstream of UBE2NL and HIST2H3PS2, particularly their influence on NF-κB activation, DNA damage repair, and chromatin remodeling. Second, therapeutic strategies targeting these genes, their interacting partners, or their secreted EVs should be verified. Inhibitors of ubiquitination pathways or EV biogenesis may provide novel treatment avenues. Third, biomarker development using UBE2NL, HIST2H3PS2, or their EV-associated cargo could improve patient risk stratification and monitoring of treatment response. Fourth, detailed EV cargo profiling via proteomic and transcriptomic analyses could reveal specific oncogenic molecules contributing to metastasis. Finally, combination therapies integrating HDAC inhibitors and EV-targeted approaches may provide synergistic anti-tumor effects. Targeting UBE2NL, HIST2H3PS2, and their EV-mediated effects may represent a promising new frontier in treating aggressive gynecologic cancers.
I would recommend uploading the TCGA data analysis pipelines/details as a supplementary file. That would be helpful for the readers and future researchers as well.
Thank you for your comment. We upload the TCGA data analysis pipelines/details as a supplementary file.
Supplementary File 1 – TCGA Data Analysis Pipeline
TCGA Data Analysis Pipeline for Ovarian and Endometrial Cancer Transcriptome and Survival Study
1. Data Sources
Ovarian Cancer (OV):
- RNA-seq read count data (Level 3, serous type) for 372 samples were downloaded from the GDC Data Portal (https://portal.gdc.cancer.gov/).
- Gene annotation reference: GENCODE version 22 from GDC Reference Files (https://gdc.cancer.gov/about-data/gdc-data-processing/gdc-reference-files).
- Clinical information: Pan-Cancer Atlas (https://gdc.cancer.gov/about-data/publications/pancanatlas).
Endometrial Cancer:
- Cohort created using GDC Data Portal (https://portal.gdc.cancer.gov/), resulting in 404 final samples after filtering.
- Clinical data included: stage, follow-up time, vital status, and recurrence.
2. Expression Normalization
TPM (Transcripts Per Million) values were calculated using the formula:
TPM_i = [(Read Count_i / Gene Length_i) / sum_j(Read Count_j / Gene Length_j)] × 10^6
3. Survival Analysis Workflow
Ovarian Cancer
Software/Packages:
- R packages: survival, survMisc, survminer (version 0.4.9), ggplot2
Step-by-step Workflow:
- Cox proportional-hazards model (coxph function) was applied to correlate TPM values with overall survival.
2. Hazard Ratio (HR) and Wald test p-values were calculated for each gene.
3. Genes were filtered with HR ≥ 2 and p-value ≤ 0.05.
4. For progression-free survival (PFS), optimal cut points were calculated using the cutp function (survMisc).
5. Samples were split into high and low expression groups based on TPM.
6. Kaplan-Meier plots were generated with ggsurvplot from survminer.
Sample Numbers Used:
- TCGA (serous type): 373 (OS), 177 (PFS)
- GEO + TCGA (all subtypes): 1656 (OS), 1435 (PFS)
Endometrial Cancer
Expression Groups:
- Patients were divided into high and low expression groups of UBE2NL and HIST2H3PS2 using TPM cutoffs determined by survminer.
Survival Subgroups Analyzed:
- Early-stage (I-II) vs. advanced-stage (III-IV)
- Cancer stage (I, II, III, IV)
- Gene high vs. low expression
4. Summary of Tools Used
Step |
Tool/Package |
Version |
TPM Calculation |
Custom R script |
– |
Cox Regression |
survival::coxph |
3.2-13 |
Optimal Cut Point |
survMisc::cutp |
0.5.5 |
Kaplan-Meier Plot |
survminer::ggsurvplot |
0.4.9 |
Visualization |
ggplot2 |
3.3.5 |
5. Notes
- All R scripts were run in R version 4.1.2 on Ubuntu 20.04 LTS.
- RNA-seq and microarray expression data were processed independently and normalized as required.

Round 2
Reviewer 1 Report
Comments and Suggestions for Authors
The authors have responded adequately to reviewers' concerns; only minor revisions are needed before publication.
- In the first sentence of the abstract, please choose either "extracellular vesicles" or "exosomes".
- There are no "a" and "b" in Figure 3.
Author Response
The authors have responded adequately to reviewers' concerns; only minor revisions are needed before publication.
- In the first sentence of the abstract, please choose either "extracellular vesicles" or "exosomes".
Response. Thank you for your comment. We revised the first sentence of the abstract to read: Extracellular vesicles (EVs) play pivotal roles in tumor progression and metastasis by mediating intercellular communication within the tumor microenvironment.
- There are no "a" and "b" in Figure 3.
Response. Thank you for your comment. We added “a” and “b” in revised Figure 3.
